# Application of Neuromuscular Blockers in Patients with ARDS in ICU: A Retrospective Study Based on the MIMIC-III Database

**DOI:** 10.3390/jcm12051878

**Published:** 2023-02-27

**Authors:** Xiaojun Pan, Jiao Liu, Sheng Zhang, Sisi Huang, Limin Chen, Xuan Shen, Dechang Chen

**Affiliations:** Department of Critical Care Medicine, Ruijin Hospital, Shanghai Jiao Tong University School of Medicine, No. 197, Ruijin 2nd Road, Shanghai 200025, China

**Keywords:** intensive care unit, ARDS, NMBAs, mortality

## Abstract

Background: Although neuromuscular blocker agents (NMBAs) are recommended by guidelines as a treatment for ARDS patients, the efficacy of NMBAs is still controversial. Our study aimed to investigate the association between cisatracurium infusion and the medium- and long-term outcomes of critically ill patients with moderate and severe ARDS. Methods: We performed a single-center, retrospective study of 485 critically ill adult patients with ARDS based on the Medical Information Mart for Intensive Care III (MIMIC-III) database. Propensity score matching (PSM) was used to match patients receiving NMBA administration with those not receiving NMBAs. The Cox proportional hazards model, Kaplan–Meier method, and subgroup analysis were used to evaluate the relationship between NMBA therapy and 28-day mortality. Results: A total of 485 moderate and severe patients with ARDS were reviewed and 86 pairs of patients were matched after PSM. NMBAs were not associated with reduced 28-day mortality (hazard ratio (HR) 1.44; 95% CI: 0.85~2.46; *p* = 0.20), 90-day mortality (HR = 1.49; 95% CI: 0.92~2.41; *p* = 0.10), 1-year mortality (HR = 1.34; 95% CI: 0.86~2.09; *p* = 0.20), or hospital mortality (HR = 1.34; 95% CI: 0.81~2.24; *p* = 0.30). However, NMBAs were associated with a prolonged duration of ventilation and the length of ICU stay. Conclusions: NMBAs were not associated with improved medium- and long-term survival and may result in some adverse clinical outcomes.

## 1. Introduction

Acute respiratory distress syndrome (ARDS) affects approximately 3 million patients globally every year [1] and accounts for approximately 10% of intensive care unit (ICU) inpatients [2,3]. Although we have made progress in our understanding of the disease, the treatment options for ARDS are still limited. The mortality of ARDS patients ranges from 40% to 60% depending on the severity of the disease, which is usually high [1,4,5,6,7].

ARDS is defined as an acute inflammatory lung injury caused by a variety of diseases, resulting in refractory hypoxemia and ultimately leading to pulmonary dysfunction, which threatens the patient’s life [2,3]. Mechanical ventilation is a key element of the treatment process for ARDS and can reduce mortality among ARDS patients [8]. Although the low-tidal-volume ventilation strategy may protect the lungs from a ventilation-related lung injury, both high pressure and a large tidal volume may occur through the spontaneous breathing effort of the patients [9].

Neuromuscular blocking agents (NMBAs) are a class of therapeutic drugs that act on the skeletal neuromuscular junction (NMJ) by inducing muscle paralysis, which can reduce the consumption of oxygen and patient–ventilator asynchrony [10]. NMBAs can improve oxygenation and decrease ventilator-induced lung injury and the work required for breathing, prevent ventilator asynchrony, and reduce airway pressure and lung stress [11]. However, NMBA therapy may not affect oxygen consumption in patients under appropriate sedation [12]. Moreover, it may result in a variety of adverse outcomes such as ICU-acquired weakness, polyneuropathy, atelectasis, muscle paralysis, etc. [10,13]. A randomized control trial (RCT) showed that continuous cisatracurium infusion can improve oxygenation in patients with ARDS [14]. Another RCT demonstrated that cisatracurium can significantly reduce the inflammatory response in ARDS patients [15]. In 2010, the ACURASYS trial recruited 339 patients and found that the early administration of cisatracurium to patients with moderate and severe ARDS improved the hospital mortality rate [11]. The PETAL trial reported that there was no significant difference in all-cause mortality on day 90 between ARDS patients who received early or continuous cisatracurium administration and those who received usual care [16].

Therefore, NMBA infusion in patients with ARDS remains controversial. The aim of this study was to evaluate the efficacy and middle- and long-term outcomes of early cisatracurium infusion in moderate and severe ARDS patients.

## 2. Materials and Methods

### 2.1. Sources of Data

Data for the study were derived from the MIMIC-III database (Medical Information Mart for Intensive Care, version 1.4). The database was approved by the Institutional Review Board (IRB) of the Massachusetts Institute of Technology (MIT). After full completion of the National Institutes of Health web-based training course and the Protecting Human Research Participants examination (NO. 35209874), permission to extract data from MIMIC-III was provided. The database is funded by the National Institutes of Health (NIH), Beth Israel Deaconess Medical Center, the Massachusetts Institute of Technology (MIT), Oxford University, and Massachusetts General Hospital (MGH), having been created by emergency doctors, intensive physicians, computer science experts, etc. The database records the data of patients admitted to the Beth Israel Deaconess Medical Center from June 2001 to October 2012. It contains more than 58,000 inpatient data points representing 38,645 adult individuals and 7875 newborns. These data are organized into tables in CSV format for research inquiries and include almost all the data of the patients during ICU treatment, such as demographic characteristics, vital signs recorded every hour, operation records, the administration time and dose of the drug used, the amount of fluid passing in and out, the results of microbiological examinations, care records, the outcomes of the patients (inpatient deaths, out-of-hospital deaths, and discharges), etc.

### 2.2. Study Cohort

We conducted a single-center, retrospective study of ARDS patients according to the Berlin definition. All the data were extracted based on the method established by Johnson et al. [13,14]. The inclusion criteria were as follows: (1) moderate and severe ARDS patients; (2) patients first admitted to ICU; (3) age ≥ 16 years old; (4) patients receiving mechanical ventilation for more than 48 h; and (5) patients receiving cisatracurium therapy. The exclusion criteria were as follows: (1) patients who died within the first 48 h; (2) removal of the endotracheal tube within 48 h; and (3) missing key data. Data were extracted from the MIMIC-III database using Structured Query Language (SQL). The following data were collected on the first day of ICU admission: weight, gender, age, admission type, ethnicity (White, Hispanic, Black, or Other), mechanical ventilation, use of NMBAs and vasopressors, renal replacement therapy (RRT), ARDS severity, simplified acute physiology score II (SAPS II) and sequential organ failure assessment (SOFA) score, heart rate, saturation of pulse oxygen (SPO2), respiratory rate, positive end-expiratory pressure (PEEP), and comorbidities. The definitions of moderate and severe ARDS were in accordance with the Berlin definition.

### 2.3. Endpoints

The primary endpoint was the 28-day mortality. The secondary endpoints were the 90-day mortality, 1-year mortality, hospital mortality, length of stay in the ICU and hospital, and ventilation duration. Moreover, we extracted the vital signs (including the heart rate, blood pressure, body temperature, and SpO2), respiratory mechanic indicators (including the tidal volume, plateau pressure, peak inspiratory pressure, PEEP, respiratory rate, and PaO2/FiO2 (P/F) ratio), Ramsay sedation scores (RASS), and total amount of fluid input and urine output of the patients from the first day of hospital admission to the seventh day.

### 2.4. Statistical Analysis

Continuous variables are summarized as the mean and standard deviation or median and interquartile range according to the data distribution, and categorical variables are presented as numbers and percentages. The Shapiro–Wilk test was used to test for a normal distribution. A Wilcoxon rank-sum test, Student’s *t*-test, or Chi-square test was performed to compare the differences between groups where appropriate. The Cox hazards model was conducted to evaluate the difference in mortality outcomes between the two groups and the confounding variables were defined according to a *p*-value < 0.05 based on univariate analysis and clinical expert judgment. Kaplan–Meier curves were created for the pre-matched and matched cohorts to assess the survival of the NMBA and non-NMBA groups.

To control the confounding factors between the two groups, propensity-score matching (PSM) was used. The propensity score of an individual was determined based on the given covariates of age, gender, ethnicity, admission type, SOFA and SAPS II scores, ARDS severity, heart rate, respiratory rate, RASS score, first-day use of vasopressors, ventilation and RRT, chronic disease of the liver, chronic obstructive pulmonary disease (COPD), chronic heart failure (CHF), and malignancy using a generalized linear model. We used random forest imputation to process the missing data before PSM. When the missing data amounted to less than 5%, random forest was performed using the “randomForest” package in R. Patients were matched in a 1:1 ratio using the nearest neighbor algorithm with a caliper of 0.2. After matching, the standardized mean differences (SMDs) between the two groups were calculated. Statistical significance was considered to be indicated by a two-sided *p* < 0.05. All the statistical analyses mentioned above were performed using RStudio (version 4.0.5).

## 3. Results

### 3.1. Baseline Characteristics 

After reviewing 61,532 subjects from the MIMIC-III database, we identified ARDS in 1349 subjects according to the Berlin definition, and 485 patients were enrolled after the application of the exclusion criteria (Figure 1). A total of 115 patients (23.71%) received NMBA therapy and 370 (76.29%) did not, as shown in Table 1. There were no significant differences in weight, gender, admission type, ethnicity, first-day use of ventilation and RRT, the SAPS II score, CHF, AFIB, CAD, malignancy, stroke, and chronic disease of the liver or renal between the two groups. The most common comorbidities were chronic heart failure and COPD, which were observed at lower frequencies in the NMBA group than in the non-NMBA group. After PSM, 86 patients who received NMBAs were matched with 86 patients who did not. The baseline was well balanced between the two groups (shown in Table 2 and Appendix A).

### 3.2. Relationship between NMBAs and Outcomes

In our study, the 28-day mortality, 90-day mortality, and 1-year mortality were 29.48%, 35.05%, and 43.09%, respectively. The results of the pre-matched cohort showed that the 28-day mortality (HR = 1.62; 95% CI: 1.14–2.30; *p* < 0.01), 90-day mortality (HR = 1.58; 95% CI: 1.14–2.19; *p* < 0.01), 1-year mortality (HR = 1.40; 95% CI: 1.04–1.90; *p* = 0.03), and hospital mortality (HR = 1.41; 95% CI: 0.99–2.00; *p* = 0.06) were associated with NMBA therapy in the original cohort. After being adjusted for the confounders (including gender, age, SOFA, SAPS II, ethnicity, ARDS severity, chronic disease of the liver, malignancy, and respiratory rate) with two COX models, NMBAs were still associated with the 28-day, 90-day, or 1-year mortality (Table 3). The median lengths of hospital stay and ICU stay were 17.09 and 10.98 days, respectively, and the median duration of ventilation was 7.59 days (Table 4). The duration of ICU stay and ventilation were longer among patients who received NMBA therapy.

After PSM, NMBA therapy use was not associated with a reduced 28-day, 90-day, 1-year, or hospital mortality in the matched cohort (Table 3). Moreover, NMBA therapy was not associated with the 28-day, 90-day, 1-year, or hospital mortality after adjusting for the possible confounding factors in the matched cohort (Table 3). However, the ventilation duration and ICU stay were 8.72 and 11.22 days, which were prolonged by NMBA administration (Table 4). Kaplan–Meier survival curves were plotted to evaluate the effect of NMBA treatment using the log-rank test, and the results are shown in Figure 2. The 28-day, 90-day, and 1-year mortality were higher in the NMBA group in the original cohort (*p* < 0.01, *p* < 0.01, *p* = 0.03). However, there was no difference in the 28-day, 90-day, or 1-year mortality between the groups in the matched cohort (*p* = 0.84, *p* = 0.95, *p* = 0.78).

The univariate COX analysis results of the 28-day mortality are shown in Appendix A. Age, the SAPSII and SOFA scores, ARDS severity, comorbidities associated with the liver and malignancy, body temperature, and respiratory rate were the risk factors for 28-day mortality. The vital signs, respiratory mechanic indicators, RASS scores, and total amount of fluid input and urine output of the patients from the first day of admission to the ICU to the seventh day are shown in Appendix A.

### 3.3. Subgroup Analysis

The results of the subgroup analysis of the 28-day mortality are shown in Figure 3. There were no differences in NMBA treatment between the subgroups. 

## 4. Discussion

NMBAs are used in 25–45% of ARDS patients through either intermittent or continuous infusion [17]. Cisatracurium is a competitive antagonist of the nicotinic acetylcholine receptors that prevents acetylcholine from binding to the receptors in order to induce reversible muscular paresis. It undergoes Hofmann elimination, which means that its metabolism does not depend on renal or hepatic function; hence, it is preferred for critically ill patients [18]. However, the data used to evaluate the efficacy of NMBAs in ARDS patients are inconsistent. The ACURASYS study showed that the early administration of neuromuscular blocking agents improved the 90-day survival rate and decreased the duration of mechanical ventilation [11]. Nevertheless, the recent PETAL trial found that early therapy with NMBAs was not significantly associated with 90-day mortality [16]. A meta-analysis showed that NMBA therapy may be beneficial for short-term mortality among patients with ARDS but not for mid- or long-term mortality [19]. Herein, we retrospectively reviewed a cohort of 485 ARDS patients from the MIMIC-III database and demonstrated that NMBAs were not associated with an increased risk of 28-day, 90-day, 1-year, or hospital mortality but may prolong the ventilation duration and length of ICU stay.

Our study showed different results from the ACURASYS trial, mainly because we used the Berlin definition, which is in contrast to the definition of the American–European Consensus Conference used in the ACURASYS trial but is the same as the definition used in the PETAL trial. Thus, there is slight heterogeneity between the population of our study and that of the ACURASYS trial. The pathophysiological process of ARDS is divided into three stages, the exudative, repaired, and proliferative phases [20]. NMBAs may also inhibit the release of inflammatory factors (IL-1β, IL-6, and IL-8, etc.) and improve the outcomes of patients in the early stage of ARDS [15,21,22]. NMBAs improved the mechanical compliance of the chest wall and induced a change in the ventilation/perfusion ratio, which could be responsible for improvements in gas exchange and oxygenation [19]. Gainnier et al. showed a significant benefit of NMBA therapy in influencing the PaO2/FiO2 ratio [14], whereas the ACURASYS study showed that the PaO2/FiO2 ratio was higher on day 7 in patients receiving NMBAs [11]. Furthermore, an increase in thoracic–pulmonary compliance in ARDS patients can increase their functional residual capacity (FRC) and decrease the degree of intrapulmonary shunt [23]. Moreover, NMBA administration improved asynchrony, which contributed to patient comfort, rendered ventilation more effective, decreased the airway pressure and work required for breathing, and prevented muscle fatigue [11,24]. Tidal volumes can be closely regulated with NMBA therapy, thus decreasing the barotrauma and volutrauma caused by the overinflation of the alveoli, which may minimize the manifestations of ventilator-induced lung injury (VILI) [11]. There are inherent risks of NMBA therapy for ICU patients following the discontinuation of neuromuscular blocking agents such as ICU-acquired weakness (ICUAW), prolonged paralysis, the development of critical illness myopathy, polyneuropathy, etc. [25]. Patients who were paralyzed and subjected to NMBA administration underwent more serious adverse events such as hypoxemia and hypercarbia, causing cardiopulmonary collapse [26]. More seriously, NMBAs led to the inhibition of the cough reflex, which hindered secretion clearance, and thus may prolong the ventilation duration and length of ICU stay. Moreover, NMBAs have very complex interactions with other drugs, such as corticosteroids, beta-blockers, calcium channel blockers, vancomycin, clindamycin, and so on, causing even more alterations in the pH and electrolyte levels [27]. Therefore, NMBAs may not result in clinical benefits due to their side effects after the exudative stage [28]. The present study did not exclude patients who used NMBAS for more than 48 h. Long-term NMBA infusion is associated with muscle paralysis [29] and ICU-acquired weakness, which may increase mortality among critically ill patients [30,31]. A prolonged length of ICU stay and mechanical ventilation duration were associated with higher mortality [32,33]. Thus, the inclusion of patients who received long-term NMBA therapy may have resulted in negative results in this study.

Another important factor may be differences in the sedation strategy. In patients who have received NMBA therapy, deep sedation may result in higher mortality and a prolonged duration of extubation [34]. The early deep sedation level is associated with higher mortality in critically ill patients who have received mechanical ventilation [34,35,36,37], whereas a light sedation strategy may improve the clinical outcomes of mechanically ventilated patients in the early stage [35,36]. Although the RASS scores on the first day of admission to the ICU were carefully propensity-score matched between the two groups, it is possible that the patients who underwent NMBA infusion were more deeply sedated than the patients who did receive NMBAs on the second day (following the first day of admission). The sedation level is associated with the prognosis of patients with ARDS [28].

## 5. Limitations

Most notably, the MIMIC-III database used in our study only contains the data of critically ill patients admitted between 2001 and 2012. Secondly, the different treatment strategies for critically ill patients, including ventilation strategies, nutritional support, and fluid management, may have influenced the outcomes of the ARDS patients. Thirdly, our study had a single-center, retrospective design; thus, the results of the present study still require further validation using external datasets. Despite our careful propensity-score matching, residual confounding factors cannot be fully excluded. Therefore, the risk of confounding factors should be taken into account when interpreting the results.

## 6. Conclusions

The use of NMBAs was not associated with reduced 28-day or 90-day mortality and may prolong the duration of ventilation and length of ICU stay. Due to their many side effects, we should use NMBAs with caution.

## Figures and Tables

**Figure 1 jcm-12-01878-f001:**
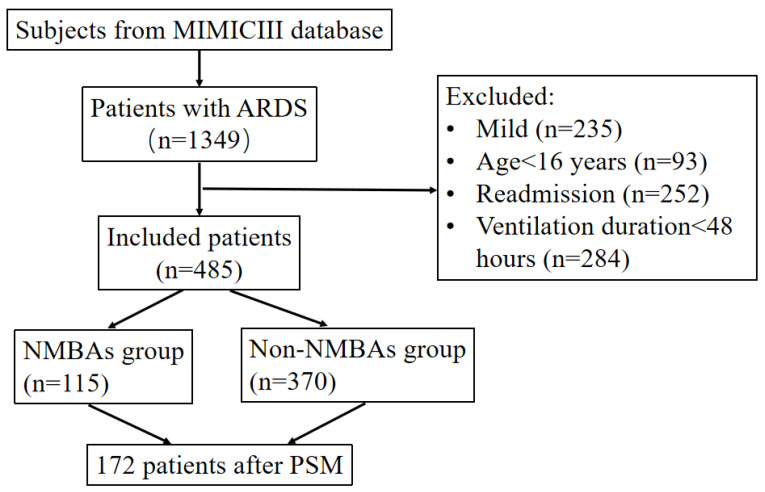
Flowchart of included patients. MIMIC-III: Multiparameter Intelligent Monitoring in Intensive Care Database III; ICU: intensive care unit; PSM: propensity-score matching.

**Figure 2 jcm-12-01878-f002:**
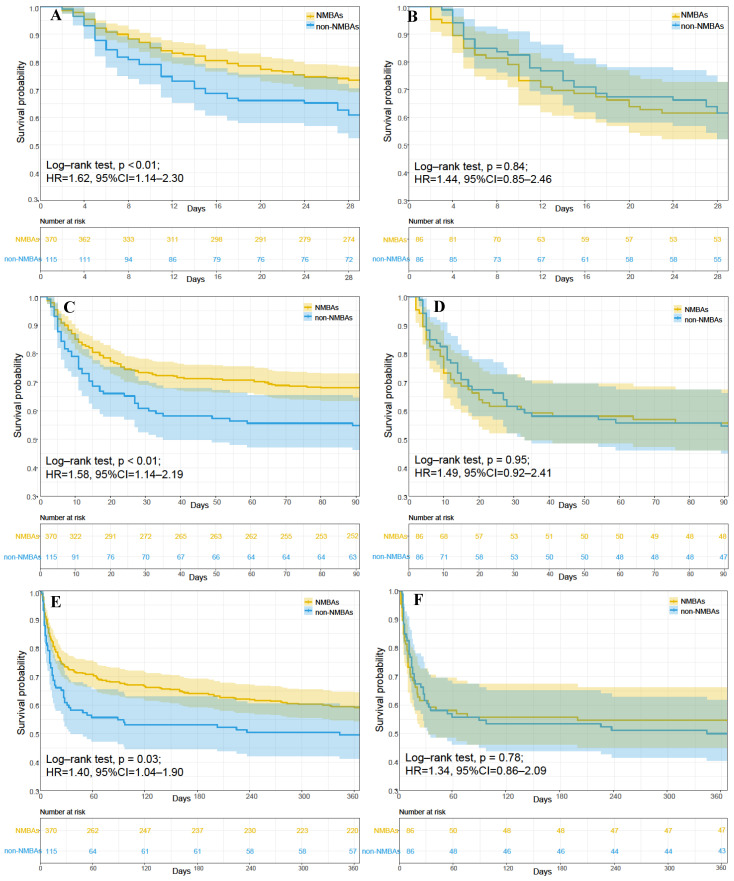
Survival analysis of NMBA and non-NMBA groups. Kaplan–Meier survival curves for the 28-day (**A**,**B**), 90-day (**C**,**D**), and 1-year (**E**,**F**) mortality among all patients are shown. Kaplan–Meier survival curves for pre-matched cohort (**A**,**C**,**E**) and matched cohort (**B**,**D**,**F**).

**Figure 3 jcm-12-01878-f003:**
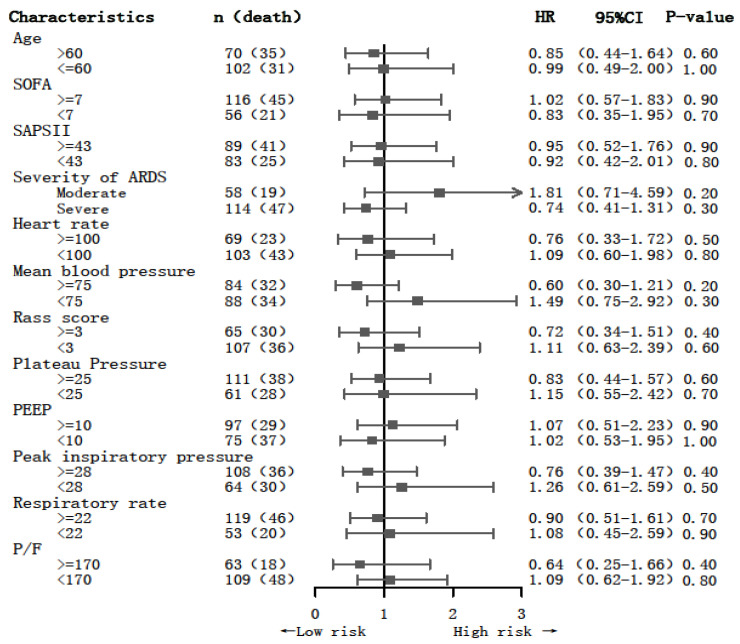
The association between NMBA administration and 28-day mortality in the subgroups.

**Table 1 jcm-12-01878-t001:** Baseline characteristics of the original cohort.

	All (n = 485)	Non-NMBAs (n = 370)	NMBAs (n = 115)	*p*-Value
Weight	80.00 [68.45, 95.00]	80.20 [68.00, 95.00]	80.00 [69.15, 94.95]	0.90
Gender (%)				0.05
Male	284 (58.56)	207 (55.95)	77 (66.96)	
Female	201 (41.44)	163 (44.05)	38 (33.04)	
Age (years)	58.52 [46.04, 72.22]	59.85 [47.02, 75.20]	56.47 [40.64, 66.45]	0.01
Admission type (%)				0.42
Elective	31 (6.39)	22 (5.95)	9 (7.83)	
Emergency	432 (89.07)	329 (88.92)	103 (89.57)	
Urgent	22 (4.54)	19 (5.14)	3 (2.61)	
Ethnicity (%)				0.96
White	307 (63.30)	236 (63.78)	71 (61.74)	
Hispanic	17 (3.51)	13 (3.51)	4 (3.48)	
Black	31 (6.39)	24 (6.49)	7 (6.09)	
Other	130 (26.80)	97 (26.22)	33 (28.70)	
Mechanical ventilation (%)	426 (87.84)	323 (87.30)	103 (89.57)	0.63
Vasopressors (%)	249 (51.34)	173 (46.76)	76 (66.09)	<0.01
RRT (%)	30 (6.19)	20 (5.41)	10 (8.70)	0.29
ARDS severity (%)				<0.01
Moderate	250 (51.55)	214 (57.84)	36 (31.30)	
Severe	235 (48.45)	156 (42.16)	79 (68.70)	
SAPS II	43.00 [33.00, 54.00]	43.00 [33.00, 53.00]	44.00 [34.50, 59.00]	0.28
SOFA	7.00 [5.00, 10.00]	7.00 [5.00, 9.00]	9.00 [6.00, 12.00]	<0.01
Heart rate (bpm)	93.27 (17.94)	92.12 (16.99)	96.99 (20.35)	0.01
SpO2	96.73 [95.43, 97.89]	97.03 [95.70, 98.08]	96.08 [94.67, 97.40]	<0.01
Respiratory rate (bpm)	22.35 (4.93)	21.64 (4.85)	24.61 (4.52)	<0.01
PEEP	8.78 [5.84, 11.22]	8.33 [5.00, 10.24]	11.70 [8.57, 14.94]	<0.01
RASS score	−1.20 [−1.44, −0.83]	−1.07 [−1.20, −0.75]	−1.72 [−2.30, −1.20]	<0.01
Co-morbidities (%)				
CHF	171 (35.26)	142 (37.40)	29 (25.22)	0.08
AFIB	115 (23.71)	92 (24.86)	23 (20.00)	0.34
CAD	44 (9.07)	35 (9.46)	9 (7.83)	0.73
Malignancy	85 (17.53)	71 (19.19)	14 (12.17)	0.11
Kidney	36 (7.42)	28 (7.57)	8 (6.96)	>0.99
Liver	31 (6.39)	24 (6.49)	7 (6.09)	>0.99
COPD	66 (13.60)	60 (16.22)	6 (5.22)	0.01
Stroke	44 (9.07)	34 (9.19)	10 (8.70)	>0.99

Abbreviations: NMBAs, neuromuscular blocking agents. ARDS, acute respiratory distress syndrome. RRT, renal replacement therapy. SAPS II, simplified acute physiology score II. SOFA, sequential organ failure assessment. PEEP, positive end-expiratory pressure. RASS sore, Richmond agitation–sedation scale score. CHF, chronic heart failure. AFIB, atrial fibrillation. CAD, coronary artery disease. COPD, chronic obstructive pulmonary disease. bpm, beats per minute. All covariates were reported as the mean (standard deviation) and median (IQR). Mechanical ventilation, vasopressors, and RRT were received on the first day of therapy. All data were extracted in the first 24 h of ICU admission.

**Table 2 jcm-12-01878-t002:** Baseline characteristics of the matched cohort.

	Matched Cohort		
	Non-NMBAs	NMBAs	SMD
n	86	86	
Gender (%)			0.12
Male	58 (67.44)	53 (61.63)	
Female	28 (32.56)	33 (38.37)	
Age (years)	52.69 (19.88)	54.45 (16.28)	0.10
Admission type (%)			0.14
Elective	5 (5.81)	7 (8.14)	
Emergency	76 (88.37)	76 (88.37)	
Urgent	5 (5.81)	3 (3.49)	
Ethnicity (%)			0.08
White	52 (60.47)	54 (62.79)	
Hispanic	5 (5.81)	4 (4.65)	
Black	5 (5.81)	4 (4.65)	
Other	24 (27.91)	24 (27.91)	
Ventilation (%)	77 (89.53)	77 (89.53)	<0.01
RRT (%)	6 (6.98)	7 (8.14)	0.04
Vasopressors (%)	57 (66.28)	51 (59.30)	0.15
ARDS severity (%)			<0.01
Moderate	29 (33.72)	29 (33.72)	
Severe	57 (66.28)	57 (66.28)	
SAPS II	45.00 (14.68)	45.62 (16.83)	0.04
SOFA	8.65 (3.60)	8.63 (3.85)	<0.01
Heart rate (bpm)	94.52 (18.08)	95.94 (19.27)	0.08
Respiratory rate (bpm)	24.49 (4.77)	24.21 (4.24)	0.06
RASS	2.69 (0.62)	2.65 (0.48)	0.07
Co-morbidities (%)			
CHF	24 (27.91)	21 (24.42)	0.08
Renal			
Liver	7 (8.14)	6 (6.98)	0.04
COPD	3 (3.49)	5 (5.81)	0.11
Stroke	9 (10.47)	9 (10.47)	<0.01

Abbreviation: SMD, standardized mean difference. All covariates are reported as the mean and standard deviation.

**Table 3 jcm-12-01878-t003:** Outcomes of NMBAs and non-NMBA patients and sensitivity analysis.

	HR	Low 95% CI	High 95% CI	*p*-Value
Pre-matched cohort				
28-day mortality	1.62	1.14	2.30	<0.01
Adjusted model I	1.78	1.23	2.56	<0.01
Adjusted model II	1.39	0.94	2.04	<0.01
90-day mortality	1.58	1.14	2.19	<0.01
Adjusted model I	1.75	1.25	2.45	<0.01
Adjusted model II	1.49	1.03	2.14	<0.01
One-year mortality	1.40	1.04	1.90	0.03
Adjusted model I	1.41	1.03	1.92	<0.01
Adjusted model II	1.39	1.00	1.95	<0.01
Hospital mortality	1.41	0.99	2.00	0.06
Adjusted model I	1.60	1.11	2.30	<0.01
Adjusted model II	1.32	0.90	1.95	<0.01
Matched cohort				
28-day mortality	1.44	0.85	2.46	0.20
Adjusted model I	1.39	0.81	2.39	0.23
Adjusted model II	1.47	0.84	2.56	0.17
90-day mortality	1.49	0.92	2.41	0.10
Adjusted model I	1.54	0.94	2.54	0.09
Adjusted model II	1.61	0.97	2.67	0.06
One-year mortality	1.34	0.86	2.09	0.20
Adjusted model I	1.34	0.85	2.10	0.20
Adjusted model II	1.41	0.89	2.22	0.15
Hospital mortality	1.34	0.81	2.24	0.30
Adjusted model I	1.39	0.83	2.32	0.21
Adjusted model II	1.48	0.87	2.52	0.15

Abbreviations: CI, confidence interval. HR, hazard ratio; All models were obtained by Cox proportional hazards model analysis of the relationship between NMBA therapy and all-cause mortality. Model I was adjusted for gender, age, admission type, and ethnicity. Model II was adjusted for gender, age, SOFA, SAPS II, ethnicity, ARDS severity, chronic disease of the liver, malignancy, and respiratory rate.

**Table 4 jcm-12-01878-t004:** Other outcomes.

	Overall	Non-NMBAs	NMBAs	*p*-Value
Pre-matched cohort	N = 485	N = 370	N = 115	
Length of hospital stay (days)	17.09 [10.11, 24.90]	16.80 [10.12, 23.76]	18.09 [9.48, 29.80]	0.21
Length of ICU stay (days)	10.98 [6.14, 18.76]	10.20 [5.96, 16.18]	14.92 [7.29, 26.79]	<0.01
Duration of ventilation (days)	7.59 [4.42, 14.00]	7.12 [4.17, 11.97]	12.29 [5.53, 20.33]	<0.01
Matched cohort	N = 172	N = 86	N = 86	
Length of hospital stay (days)	17.15 [9.62, 26.70]	14.92 [9.43, 22.02]	18.05 [11.56, 28.95]	0.06
Length of ICU stay (days)	11.22 [6.02, 19.94]	9.37 [5.47, 12.86]	14.67 [7.96, 26.22]	<0.01
Duration of ventilation (days)	8.72 [4.42, 15.59]	6.40 [3.40, 10.48]	12.38 [5.55, 19.77]	<0.01

Data are represented by median (IQR).

## Data Availability

The datasets used in the present study are available from the authors upon reasonable request.

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
