# Peer review of "Application of Neuromuscular Blockers in Patients with ARDS in ICU: A Retrospective Study Based on the MIMIC-III Database"

_jcm, 2023, doi:10.3390/jcm12051878_

Round 1
Reviewer 1 Report
The manuscript “Application of neuromuscular blockers in patients with ARDS in ICU: a retrospective study from MIMIC database” is a retrospective investigation of patients entered into a database. The study investigated the association between administration of neuromuscular blockers and outcomes in patients with moderate to severe ARDS. The study is carefully designed and the results comprehensively described. Particularly, the conclusion that “With so many side effects, we should use NMBAs with caution” is pertinent. However, the manuscript would be further improved by including the following comments:
Overall, the text would benefit from language editing to be easier to read. For example, line 13 reads: in critically ill patients with moderate to severe ARDS patients. Please revise.
Line 30: References before the full stop.
Line 80: ethnicity (White, Black or Other)
In table 1, you have white, black, Hispanic and other.
Please be consistent.
Further comment to table 1: Does the p-value of 0.01 for age mean that the patients given NMBAs were younger? Could this not affect the outcome?
Line 96: The Chi-square test or Student's t-test was appropriately performed to compare the differences between groups. Please let the reviewers decide if the use of different statistical tests was appropriate or not.
Line 116: After reviewing 61532 subjects from MIMIC-III, we identified ARDS in 1349 subjects selected through the Berlin Definition, 485 eligible patients were enrolled according to the exclusion criteria. 115 patients (23.71%) received NMBAs therapy and 370 (76.29%) did not, as shown in Table 1.
1: What do you mean by the statement that 485 patients were enrolled according to the exclusion criteria. They should have been enrolled according to the inclusion criteria or excluded based on exclusion criteria.
2: How could only 1349 out of over 61000 patients be included? Why were the rest of the patients excluded?
Line 122: There were 250 (51.55) moderate ARDS patients and 235 (48.45) severe, respectively.
What was the definition of moderate and severe ARDS, respectively?
Line 153: After PS matching, 86 patients who received NMBAs were matched with 86 patients who did not.
What is PS matching?
Author Response
We thank for the comments very much, these suggestions really help us a lot. We have carefully reviewed the comments and have revised the manuscript accordingly. We tried our best to amend the relevant part in manuscript according to your advice. We feel sorry for our poor writings, many grammatical or typographical errors have been revised in the revision. Our responses are given in a point-by-point manner below, changes to the manuscript are shown in red. If there are any other modifications we could make, we would like very much to modify it and we really appreciate your help. Thank you very much for your help. Please see the attachment.
Reviewer #1:
The manuscript “Application of neuromuscular blockers in patients with ARDS in ICU: a retrospective study from MIMIC database” is a retrospective investigation of patients entered into a database. The study investigated the association between administration of neuromuscular blockers and outcomes in patients with moderate to severe ARDS. The study is carefully designed and the results comprehensively described. Particularly, the conclusion that “With so many side effects, we should use NMBAs with caution” is pertinent. However, the manuscript would be further improved by including the following comments:
Overall, the text would benefit from language editing to be easier to read. For example, line 13 reads: in critically ill patients with moderate to severe ARDS patients. Please revise.
Reply: Thank you for your comments. Many grammatical or typographical errors have been revised in the revision. We have removed the last word “patients” of the sentence in the revised manuscript (line 13).
Line 30: References before the full stop.
Reply: Thank you for your suggestion. We have revised all the references before the full stop (line 31).
Line 80: ethnicity (White, Black or Other)
In table 1, you have white, black, Hispanic and other.
Please be consistent.
Reply: Thank you for your suggestion. We are sorry for our carelessness. we have revised the description in the revised manuscript (line 111).
Further comment to table 1: Does the p-value of 0.01 for age mean that the patients given NMBAs were younger? Could this not affect the outcome?
Reply: Thank you for your comments. Age may be a risk factor of mortality in the previous study (J Trauma Acute Care Surg. 2019 May;86(5):844-852). However, propensity score matching (including age, gender, ethnicity, etc) was used to control the potential confounders in our study which was widespread use in retrospective study (Zhao et al. Critical Care (2020) 24:75; Wongtangman et al, Critical Care Medicine (2021) Jul 1;49(7):1137-1148; Valentine Léopold, et, al. Intensive Care Med. 2018 Jun;44(6):847-856. Alexandre Kalimouttou, et, al. Intensive Care Med. 2023 Jan;49(1):26-36.). Meanwhile, age in the two groups was well balanced in the matched cohort which was shown in Table 2. Moreover, we also adjusted potential confounders including age in the Cox proportional hazard models which was shown in Adjust mode I and Adjust mode II of Table 3.
Line 96: The Chi-square test or Student's t-test was appropriately performed to compare the differences between groups. Please let the reviewers decide if the use of different statistical tests was appropriate or not.
Reply: Thank you for your comments. Shapiro-Wilk test was used to test for normal distribution. Quantitative data will be described using the mean and standard deviation if the data are normally distributed, or will be described using median and interquartile range. Qualitative data will be described as the number of cases and proportions. The independent Student t-test or the Wilcoxon-rank-sum test will be performed for comparison of quantitative data between groups according to the data distribution. We have revised the description in line 129 to 134.
Line 116: After reviewing 61532 subjects from MIMIC-III, we identified ARDS in 1349 subjects selected through the Berlin Definition, 485 eligible patients were enrolled according to the exclusion criteria. 115 patients (23.71%) received NMBAs therapy and 370 (76.29%) did not, as shown in Table 1.
1: What do you mean by the statement that 485 patients were enrolled according to the exclusion criteria. They should have been enrolled according to the inclusion criteria or excluded based on exclusion criteria.
Reply: Thank you for your comments. We have revised the description as “485 patients were enrolled according to the inclusion criteria or excluded based on exclusion criteria” (line 153)
2: How could only 1349 out of over 61000 patients be included? Why were the rest of the patients excluded?
Reply: Thank you for your comments. Another study on ARDS patients from MIMIC III database enrolled 1341 patients (Crit Care. 2021 Apr 20;25(1):150). Therefore, the number of ARDS patients were near to our study from this database according to the Berlin definition. Moreover, data was extracted from the MIMIC database by Structured Query Language code which was provided by the official (Critical care medicine. 2004;32(1):113-9; Crit Care Med. 2016;44(11):2079-103; Intensive Care Med. 2018 Nov;44(11):1914-1922). Moreover, we just included patients who was diagnosed ARDS within 48 hours after ICU admission. Therefore, the proportion of ARDS patients was low in the present study.
Line 122: There were 250 (51.55) moderate ARDS patients and 235 (48.45) severe, respectively.
What was the definition of moderate and severe ARDS, respectively?
Reply: Thank you for your comments. The definition of moderate and severe ARDS was according to the Berlin definition. Acute onset (within 7 days of new or worsening respiratory symptoms); Bilateral radiographical opacities that are not fully explained by effusion, atelectasis, or masses; Arterial hypoxaemia defined by thresholds: Mild: 200 < PaO2/FiO2 ratio ≤300 mm Hg, on CPAP or PEEP ≥5 cm H2O (observed mortality 27%); Moderate: 100 < PaO2/FiO2 ratio ≤200 mm Hg, on PEEP ≥5 cm H2O (observed mortality 32%); Severe: PaO2/FiO2 ratio ≤100 mm Hg, on PEEP ≥5 cm H2O (observed mortality 45%) (Lancet. 2021 Aug 14;398(10300):622-637.)
Line 153: After PS matching, 86 patients who received NMBAs were matched with 86 patients who did not.
What is PS matching?
Reply: Thank you for your comments. We are sorry for the inaccurate description. We have revised “PS matching” to “PSM (propensity score matching )” (line 190).

Reviewer 2 Report
1.- In the Table I (Characteristics of Participants).
I can read that the 2 groups are not similar, because.
a.- Age, p = 0.01
b.- Vasopressors p = 0.001
c.- ARSD Severity, the most important p <0.001
In the group that received NMBAs, there are more % of patients with severe condition.
d.- SOFA, very important p <0.01
Worse scores in the group that received NMBAs.
e.- Heart Rate p < 0.01
f.- SpO2. p < 0.01
g.- Respiratory Rate p <0.01
h.- COPD p 0.012
Ideally, when we want to compare 2 groups (with exposition and without exposition) in a COHORT Study.
The characteristics of the participants must be similar, except in the exposition.
And the p must be > 0.05
But here, these 2 groups are different, because have many p < 0.05
So those groups are not comparable.
There is a big bias in the research.
2.- In the figure: "Matched Cohort"
In the last column, you have written:
SMD = standardized mean difference ?
But, by example: Ventilation % Non NMBAs 77 (89.5)
Ventilation % NMBAs 77 (89.5) SMD <0.001
Is it correct?
3.- In the article in the results about Mortality:
28 day mortality HR: 1.62 (C.I 1.14 - 2.30) p = 0.008 Ok.
90 day mortality HR : 1.58 (CI 1.14 - 2.19) p = 0.08 p without statistical significance.
One year mortality HR: 1.4O CI 1.04 - 1.90 p = 0.03 Ok
Hospital Mortality HR: 1.41. CI 0.99- 2.00 (1 it is included in C.I ) so there is not accuracy and p = 0.06 so there is not statistical significance.
4.- Nevertheless, in the Abstract, there are others results in relation to Mortality:
a.- In all results it is included 1 in the C.I, so there is not accuracy.
b.- And p values don´t have statistical significance.
5.- I recommend to read:
a.-Shuai Shao, Hanyujie Kang, Zhaohui Tong
Early neuromuscular blocking agents for adults with acute respiratory distress syndrome: a systematic review, meta-analysis, meta- regression.
BMJ Open volume 10, Issue 11, 2020.
b.- Moon Seong Baek, Jong Ho Kim, Yaeji Lim, Young Suk Kwon
Neuromuscular blockade in mechanically ventilated pneumonia patients with moderate to severe hypoxemia: A multicenter retrospective study.
Plos One 17(12): e0277503, 2022.
c.- Makowski, Courtney; Lizza , Bryan.
Timing of administration of Neuromuscular Blocking Agents in Acute Respiratory Distress Syndrome.
Critical Care Medicine 47(1):p 13, January 2019.
Author Response
We thank you for your comments very much, these suggestions really help us a lot. We have carefully reviewed the comments and have revised the manuscript accordingly. we tried our best to amend the relevant part in manuscript according to your advice. We feel sorry for our poor writings, many grammatical or typographical errors have been revised in the revision. Our responses are given in a point-by-point manner below, changes to the manuscript are shown in red. If there are any other modifications we could make, we would like very much to modify them and we really appreciate your help. Thank you very much for your help. Please see the attachment.
1.- In the Table I (Characteristics of Participants).
I can read that the 2 groups are not similar, because.
a.- Age, p = 0.01
b.- Vasopressors p = 0.001
c.- ARSD Severity, the most important p <0.001
In the group that received NMBAs, there are more % of patients with severe condition.
d.- SOFA, very important p <0.01
Worse scores in the group that received NMBAs.
e.- Heart Rate p < 0.01
f.- SpO2. p < 0.01
g.- Respiratory Rate p <0.01
h.- COPD p 0.012
Ideally, when we want to compare 2 groups (with exposition and without exposition) in a COHORT Study.
The characteristics of the participants must be similar, except in the exposition.
And the p must be > 0.05
But here, these 2 groups are different, because have many p < 0.05
So those groups are not comparable.
There is a big bias in the research.
Reply: Thank you for your comments. In the retrospective study, there was no randomized allocation process, so the baselines of the two cohorts may be different which was common in the retrospective study. The unbalance of baseline may be a risk factor of mortality. However, propensity score matching (including age, gender, ethnicity, etc) was used to control the potential confounders in the present study which was widespread use in retrospective study (Zhao et al. Critical Care (2020) 24:75; Wongtangman et al, Critical Care Medicine (2021) Jul 1;49(7):1137-1148; Valentine Léopold, et, al. Intensive Care Med. 2018 Jun;44(6):847-856. Alexandre Kalimouttou, et, al. Intensive Care Med. 2023 Jan;49(1):26-36.). Meanwhile, the baseline in the two groups was well balanced in the matched cohort which was shown in Table 2. Moreover, to Minimize the bias we also adjusted potential confounders in the Cox proportional hazard models in original and matched cohort which was shown in Adjust mode I and Adjust mode II of Table 3.
2.- In the figure: "Matched Cohort"
In the last column, you have written:
SMD = standardized mean difference ?
But, by example: Ventilation % Non NMBAs 77 (89.5)
Ventilation % NMBAs 77 (89.5) SMD <0.001
Is it correct?
Reply: Thank you for your comments. Yes, it is correct. Standardized mean difference (SMD) <0.001 often means the well balance between the two groups (Wongtangman et al, Critical Care Medicine (2021) Jul 1;49(7):1137-1148).
3.- In the article in the results about Mortality:
28 day mortality HR: 1.62 (C.I 1.14 - 2.30) p = 0.008 Ok.
90 day mortality HR : 1.58 (CI 1.14 - 2.19) p = 0.08 p without statistical significance.
One year mortality HR: 1.4O CI 1.04 - 1.90 p = 0.03 Ok
Hospital Mortality HR: 1.41. CI 0.99- 2.00 (1 it is included in C.I ) so there is not accuracy and p = 0.06 so there is not statistical significance.
Reply: Thank you for your comments. Yes, the hospital mortality is no significant different in the original cohort which may due to the imbalance in baseline characteristics. Therefore, propensity score matching was performed to balance the baseline characteristics and control the potential confounders. The main analysis of primary outcome was based on the matched cohort in our study according to previous study (Zhao et al. Critical Care (2020) 24:75; Wongtangman et al, Critical Care Medicine (2021) Jul 1;49(7):1137-1148; Valentine Léopold, et, al. Intensive Care Med. 2018 Jun;44(6):847-856. Alexandre Kalimouttou, et, al. Intensive Care Med. 2023 Jan;49(1):26-36). The conclusion was also base on the results of matched cohort.
4.- Nevertheless, in the Abstract, there are others results in relation to Mortality:
a.- In all results it is included 1 in the C.I, so there is not accuracy.
b.- And p values don´t have statistical significance.
Reply: Thank you for your comments. The main analysis of primary outcome was based on the matched cohort which was shown in Matched cohort part of Table 3. Therefore, we use the results after propensity score matching analysis. The CI include 1 as you say and p values don´t have statistical significance, so we come to a conclusion of NMBAs was not associated with improved long-term survival and might bring out some adverse clinical outcome
5.- I recommend to read:
a.-Shuai Shao, Hanyujie Kang, Zhaohui Tong
Early neuromuscular blocking agents for adults with acute respiratory distress syndrome: a systematic review, meta-analysis, meta- regression.
BMJ Open volume 10, Issue 11, 2020.
b.- Moon Seong Baek, Jong Ho Kim, Yaeji Lim, Young Suk Kwon
Neuromuscular blockade in mechanically ventilated pneumonia patients with moderate to severe hypoxemia: A multicenter retrospective study.
Plos One 17(12): e0277503, 2022.
c.- Makowski, Courtney; Lizza , Bryan.
Timing of administration of Neuromuscular Blocking Agents in Acute Respiratory Distress Syndrome.
Critical Care Medicine 47(1):p 13, January 2019.
Reply: Thank you for your comments. We have read the article you recommend and cited the article in our manuscript ( references 31 and 34). However, we can not find the article of “Makowski, Courtney; Lizza , Bryan. Timing of administration of Neuromuscular Blocking Agents in Acute Respiratory Distress Syndrome. Critical Care Medicine 47(1):p 13, January 2019.”

Reviewer 3 Report
Regarding the manuscript "Application of neuromuscular blockers in patients with ARDS in ICU: a retrospective study from MIMIC database", I would like to highlight the efforts of the authors in this research that may become relevant in clinical practice.
Introduction: appropriate, although it should be improved. Authors should focus on the information so far published on the use of NMBAs in ARDS. They should not focus so much on ARDS pathophysiology.
The information provided in lines 47 - 50 "A study involving 339 patients... and hospital discharge (15)". It can be deleted, as it is more appropriate for the Discussion section.
However, I think it might be helpful for the potential reader if the authors state here what "the aim of this study" is.
I recommend checking if the references should be placed before or after the punctuation marks and homogenize their presentation.
methods:
Authors should state in the first paragraph of this section whether an ethics committee approval has been sought for the performance of this retrospective analysis.
In addition, I consider that the manuscript should adhere to the STROBE checklist and that it should be presented as supplementary material.
The inclusion and exclusion criteria are unordered and randomly separated by periods or commas. This should be homogenized.
An exclusion criterion is "Remove the trachea within 48 hours", I think the authors meant "remove the endotracheal tube", but this should be modified.
The information about the data extracted from the database is out of order and in the same paragraph as the inclusion/exclusion criteria. I believe that this information should be better reflected and more specific (exactly what data was extracted and how was it measured?). This is critical for the reproducibility of the study.
I recommend starting the heading "Statistical analysis" with a capital letter.
Results: This section is messy and the information provided may seem unclear to the potential reader.
Authors should refer to the first paragraph "Baseline Characteristics" instead of "Basic Characteristics."
The presentation of the included patients is not clear (lines 116-118). It should say "after application of exclusion criteria" and not "according to exclusion criteria".
The information provided in lines 119-126 ("The median (interquartile range, IQR) age... beats per minute") could be deleted, since it is duplicated in Table 1.
The authors establish that there are significant differences (I understand that statistically) in terms of weight, gender, type of income, etc. However, when studying Table 1, it can be seen that p>0.05 in these variables. So I recommend reviewing this information (it can be considered a really serious statistical error).
Table 1 should make it explicit that it refers to "mechanical ventilation", not just "ventilation".
The footer of table 1 is poorly designed.
I do not understand the information provided by tables 2 and 3.
Table 3 should explain what HR means.
The order of the tables and figures is not correct: why does table 3 appear first and then table 2? In my opinion, figure 1 should be presented at the beginning of the Results section.
The information provided by Supplementary Table 1 seems interesting and promising, I consider that it should be explained in the text.
Discussion: I think this section could be improved: There are many articles on the management of NMBAs in the ARDS that the authors have not explored.
Was cisatracurium used in all patients? The information provided in lines 224-225 in which cisatracurium is compared with vecuronium is irrelevant.
I have not found information on ICUAW in the manuscript, so I do not think that much emphasis should be placed in the Discussion section on this side effect. I believe that more emphasis should be placed on the mortality factors that could have skewed the mortality differences between the two groups detected in the study.
Author Response
We thank you for your comments very much, these suggestions really help us a lot. We have carefully reviewed the comments and have revised the manuscript accordingly. we tried our best to amend the relevant part in manuscript according to your advice. We feel sorry for our poor writings, many grammatical or typographical errors have been revised in the revision. Our responses are given in a point-by-point manner below, changes to the manuscript are shown in red. If there are any other modifications we could make, we would like very much to modify them and we really appreciate your help. Thank you very much for your help. Please see the attachment
Reviewer #2:
Regarding the manuscript "Application of neuromuscular blockers in patients with ARDS in ICU: a retrospective study from MIMIC database", I would like to highlight the efforts of the authors in this research that may become relevant in clinical practice.
Introduction: appropriate, although it should be improved. Authors should focus on the information so far published on the use of NMBAs in ARDS. They should not focus so much on ARDS pathophysiology.
Reply: Thank you for your suggestion. We have deleted the description of ARDS pathophysiology and revised the introduction to focus on the information so far published on the use of NMBAs in ARDS according to your suggestion (line 36 to 66), it really help us a lot.
The information provided in lines 47 - 50 "A study involving 339 patients... and hospital discharge (15)". It can be deleted, as it is more appropriate for the Discussion section.
Reply: Thank you for your suggestion. We have removed this part to the discussion section according to your suggestion (line 72 to 78, line 249 to 254).
However, I think it might be helpful for the potential reader if the authors state here what "the aim of this study" is.
Reply: Thank you for your comments. We have added the aim of this study in the revised manuscript (line 64 t 66).
I recommend checking if the references should be placed before or after the punctuation marks and homogenize their presentation.
Reply: Thank you for your suggestion. We have revised all the references before the punctuation marks (line 31).
methods:
Authors should state in the first paragraph of this section whether an ethics committee approval has been sought for the performance of this retrospective analysis.
Reply: Thank you for your suggestion. We have added the ethics committee approval in this section (line 82 to 86).
In addition, I consider that the manuscript should adhere to the STROBE checklist and that it should be presented as supplementary material.
Reply: Thank you for your suggestion. We have checked our manuscript according to the STROBE checklist. Meanwhile, we uploaded the STROBE checklist as supplementary material according to your advise.
The inclusion and exclusion criteria are unordered and randomly separated by periods or commas. This should be homogenized.
Reply: Thank you for your comments. We are really sorry for our carelessness, we have revised the punctuation here (line 101 to 109).
An exclusion criterion is "Remove the trachea within 48 hours", I think the authors meant "remove the endotracheal tube", but this should be modified.
Reply: Thank you for your suggestion. We have revised the description here according to your suggestion (line 106).
The information about the data extracted from the database is out of order and in the same paragraph as the inclusion/exclusion criteria. I believe that this information should be better reflected and more specific (exactly what data was extracted and how was it measured?). This is critical for the reproducibility of the study.
Reply: Thank you for your suggestion. We have revised the description here in order to consistent with Table 1 (line 110 to 116).
I recommend starting the heading "Statistical analysis" with a capital letter.
Reply: Thank you for your suggestion. We have revised the letter according to your suggestion (line 127).
Results: This section is messy and the information provided may seem unclear to the potential reader.
Authors should refer to the first paragraph "Baseline Characteristics" instead of "Basic Characteristics."
Reply: Thank you for your suggestion. We have revised the title of the first paragraph (line 151).
The presentation of the included patients is not clear (lines 116-118). It should say "after application of exclusion criteria" and not "according to exclusion criteria".
Reply: Thank you for your suggestion. We have revised the description here according to your suggestion (line 153).
The information provided in lines 119-126 ("The median (interquartile range, IQR) age... beats per minute") could be deleted, since it is duplicated in Table 1.
Reply: Thank you for your comment. We have deleted this part according to your advice (line 156 to 164). It really helps us a lot.
The authors establish that there are significant differences (I understand that statistically) in terms of weight, gender, type of income, etc. However, when studying Table 1, it can be seenthat p>0.05 in these variables. So I recommend reviewing this information (it can be considered a really serious statistical error).
Reply: Thank you for your suggestion. After reviewed this information and we feel really sorry for the mistake we have made here, we have revised the description “There were no significant differences …” (line 164).
Table 1 should make it explicit that it refers to "mechanical ventilation", not just "ventilation".
The footer of table 1 is poorly designed.
Reply: Thank you for your suggestion. We have revised the "mechanical ventilation" and the footer of Table 1 according to the requirements of journal.
I do not understand the information provided by tables 2 and 3.
Reply: Thank you for your comments. To control confounding factors between the two groups, propensity score matching (PSM) was used. We are going to show the baseline was well balanced between the two groups (line 188) after PSM which was shown in Table 2 as previous study did (Wongtangman et al, Critical Care Medicine (2021) Jul 1;49(7):1137-1148). Moreover, For the sensitivity analysis, univariate Cox proportional hazard models were performed in original and matched cohort to estimate the efficacy of NMBAs infusion for 28-day, 90-day, one year and hospital mortality in three model which was adjusted by the potential confounders such as gender, age, admission typr, ethnicity, etc (as shown in Table 3).
Table 3 should explain what HR means.
Reply: Thank you for your comments. We have added the explanation of HR in the footer of table 3.
The order of the tables and figures is not correct: why does table 3 appear first and then table 2? In my opinion, figure 1 should be presented at the beginning of the Results section.
Reply: Thank you for your comments. We uploaded the table and figure in sequence. However, the system automatically changes the sequence, We'll communicate with editors to adjust the order of the table and figure according to you suggestion.
The information provided by Supplementary Table 1 seems interesting and promising, I consider that it should be explained in the text.
Reply: Thank you for your comments. We have added the descript the results of Supplementary Table 1 in the revision (line 227 to 229).
Discussion: I think this section could be improved: There are many articles on the management of NMBAs in the ARDS that the authors have not explored.
Reply: Thank you for your comments. We have re-searched the relevant literature and revised the whole discussion section according to your suggestion (line 269 to 285).
Was cisatracurium used in all patients? The information provided in lines 224-225 in which cisatracurium is compared with vecuronium is irrelevant.
Reply: Thank you for your comments. All patients in the present study were received cisatracurium therapy, we have deleted the sentence (line 308).
I have not found information on ICUAW in the manuscript, so I do not think that much emphasis should be placed in the Discussion section on this side effect. I believe that more emphasis should be placed on the mortality factors that could have skewed the mortality differences between the two groups detected in the study.
Reply: Thank you for your comments. We have deleted the side effect of ICUAW in discussion section and emphasis the potential influence factor on our result (line 292 to 320).

Round 2
Reviewer 2 Report
1.- I have read the new version of the article.
I recognize the effort that you have done.
2.- The retrospective studies always have biases in relation to prospective studies.
3.- The lapse of study is: June 2001 - October 2012.
4.- Always is recommended to research until the last years.
5.- Don´t forget to underline the word: RETROSPECTIVE Study.
6.- It´s a better version in relation to the previous one.
7.- don´t forget to underline the limitations of the study.
Author Response
Reviewer 2#:
Reply: We thank you for your comments very much, these suggestions really help us a lot. We have carefully reviewed the comments and have revised the manuscript accordingly. We tried our best to amend the relevant part in manuscript according to your advices. Our responses are given in a point-by-point manner below, changes to the manuscript are shown in red. If there are any other modifications we could make, we would like very much to modify them and we really appreciate your help. Thank you very much for your help.
1.- I have read the new version of the article.
I recognize the effort that you have done.
Reply: Thank you for your positive reply. Your valuable advices really help us a lot.
2.- The retrospective studies always have biases in relation to prospective studies.
Reply: Thank you for your comments. The retrospective studies always have biases in relation to prospective studies as you say, we have pointed out the study was a retrospective study in the title and the Method part. Moreover, we descripted this may the limitation of this study.
3.- The lapse of study is: June 2001 - October 2012.
Reply: Thank you for your comment. The database recorded the data of patients admitted to the Beth Israel Deaconess Medical Center from June 2001 to October 2012 which was mentioned in the Method part and limitation.
4.- Always is recommended to research until the last years.
Reply: Thank you for your suggestion. The latest database of MIMIC was updated to MIMIC-IV (from 2008 to 2019). However, the imaging data was still no sharing. Thus, we couldn’t get the ARDS patients according to Berlin definition. Moreover, we have point out this may be the limitation of the present study.
5.- Don´t forget to underline the word: RETROSPECTIVE Study.
Reply: Thank you for your suggestion. We have emphasized in the title, methods and the limitation in the manuscript as listed below:
1: Title. “Application of neuromuscular blockers in patients with ARDS in ICU: a retrospective study from MIMIC-III database”
2: Method. “We conducted a single-center, retrospective study of ARDS patients according to the Berlin definition.”
3: Limitation. “It was a single-center retrospective design.”
6.- It´s a better version in relation to the previous one.
Reply: Thank you for your comment.
7.- don´t forget to underline the limitations of the study.
Reply: Thank you for your suggestion. We have pointed out the possible limitation of the present study in Limitation after discussion in the manuscript (line 277 to 285).

Reviewer 3 Report
I would like to congratulate the authors for the revision of the manuscript. The authors have made all the changes suggested by the previous review. I consider that the article has improved substantially, at least from the methodological and writing point of view, and it is much easier to read.
Figure 2 may seem difficult to understand: so much information in so little space.
Author Response
Reviewer 3#:
Reply: We thank you for your comments very much, these suggestions really help us a lot. We have carefully reviewed the comments and have revised the manuscript accordingly. We tried our best to amend the relevant part in manuscript according to your advices. Our responses are given in a point-by-point manner below, changes to the manuscript are shown in red. If there are any other modifications we could make, we would like very much to modify them and we really appreciate your help. Thank you very much for your help.
I would like to congratulate the authors for the revision of the manuscript. The authors have made all the changes suggested by the previous review. I consider that the article has improved substantially, at least from the methodological and writing point of view, and it is much easier to read.
Figure 2 may seem difficult to understand: so much information in so little space.
Reply: Thank you for your comment. We have re-arranged the figures and increased the font size for easier reading by the reader. At the same time, we have supplemented the captions to make it easier for the reader to understand the meaning expressed by the figures. The figure legend was revised as listed below: “Figure 2. Survival analysis between NMBAs and non-NMBAs group. Shown are Kaplan–Meier survival curves among all the ARDS patients of 28 days (A, B), 90 days (C, D) and, one year (E, F) mortality. Kaplan-Meier survival curves in pPre-matched cohort (A, C, E) and matched cohort (B, D, F).”
